# Autochthonous *Enterococcus durans* PFMI565 and *Lactococcus lactis* subsp. *lactis* BGBU1–4 in Bio-Control of *Listeria monocytogenes* in Ultrafiltered Cheese

**DOI:** 10.3390/foods10071448

**Published:** 2021-06-22

**Authors:** Marina Ivanovic, Nemanja Mirkovic, Milica Mirkovic, Jelena Miocinovic, Ana Radulovic, Tatjana Solevic Knudsen, Zorica Radulovic

**Affiliations:** 1Faculty of Agriculture, University of Belgrade, 11080 Belgrade, Serbia; ivanovicmarina75@gmail.com (M.I.); petrusicm@agrif.bg.ac.rs (M.M.); jmiocin@agrif.bg.ac.rs (J.M.); aradulovic@agrif.bg.ac.rs (A.R.); zorica@agrif.bg.ac.rs (Z.R.); 2Institute of Chemistry, Technology and Metallurgy, University of Belgrade, 11000 Belgrade, Serbia; tsolevic@chem.bg.ac.rs

**Keywords:** lactic acid bacteria, *Listeria monocytogenes*, UF cheese, antilisterial activity

## Abstract

Nowadays, consumers are interested in cheese produced without chemical additives or high-temperature treatments, among which, protective lactic acid bacteria (LAB) cultures could play a major role. In this study, the aims were to isolate, identify and characterize antilisterial LAB from traditionally produced cheese, and utilize suitable LAB in cheese production. Among 200 isolated LAB colonies, isolate PFMI565, with the strongest antilisterial activity, was identified as *Enterococcus durans*. *E. durans* PFMI565 was sensitive to clinically important antibiotics (erytromicin, tetracycline, kanamycin, penicillin, vancomycin) and had low acidifying activity in milk. *E. durans* PFMI565 and the previously isolated bacteriocin producer, *Lactococcus lactis* subsp. *lactis* BGBU1–4, were tested for their capability to control *Listeria monocytogenes* in experimentally contaminated ultrafiltered (UF) cheeses during 35 days of storage at 4 °C. The greatest reductions of *L. monocytogenes* numbers were achieved in UF cheese made with *L. lactis* subsp. *lactis* BGBU1–4 or with the combination of *L. lactis* subsp. *lactis* BGBU1–4 and *E. durans* PFMI565. This study underlines the potential application of *E. durans* PFMI565 and *L. lactis* subsp. *lactis* BGBU1–4 in bio-control of *L. monocytogenes* in UF cheese.

## 1. Introduction

Cheeses are a significant part of human diets because of their chemical composition and high contents of vitamins, fatty acids, minerals, bioactive compounds, and probiotic bacteria [1,2]. Cheeses are classified based on several factors: type of milk used for production (whey cheese, ultrafiltration, soured milk), fat content, consistency, type of fermentation, and texture [1]. Cheeses made from ultrafiltered milk (UF cheeses), a type of soft cheese, are very popular in Serbia, and can be stored in brine or with salt added to the milk or curd during production. However, UF cheeses with pH > 4.3 and high water activity are suitable matrices for growth of the pathogen *Listeria monocytogenes* [3,4]. Although UF cheeses are manufactured using pasteurized milk, contamination of this type of cheese sometimes occurs, usually during the production process or in post-processing manipulation of cheeses [5].

Despite the fact that the incidence of *L. monocytogenes* infections accounts for a low proportion of foodborne illnesses, the high mortality rate of listeriosis (20–30%) means this pathogen is responsible for many of the fatalities linked to food [6]. The infective dose of *L. monocytogenes* in food is high, typically ˃10^4^ cfu g^−1^(mL^−1^), but in the case of immunocompromised individuals, the infective dose ranges from 10^2^–10^4^ cfu g^−1^(mL^−1^) [7,8].

According to European Union (EU) and Serbian regulations, the number of *L. monocytogenes* in ready-to-eat products has to be less than 10^2^ cfu g^−1^ (mL^−1^) of food product during the entire shelf life [9,10]. *L. monocytogenes* is ubiquitous in the environment and it is very difficult to eliminate once it is established in food production plants. Thus, elimination and control of this pathogen in foods and food plants are imperative aims in food production [11]. Contamination of food by *L. monocytogenes* can be decreased by controlling the pathogen in the environment or during food production [12].

During food production, high heat or chemical treatments are very useful in control of *L. monocytogenes*, although these methods can cause some changes in the sensory and nutritional qualities of the food [13]. Apart from these traditional methods of *L. monocytogenes* control in food production, a potentially different approach could be using lactic acid bacteria (LAB) with antilisterial activity. LAB can produce different antimicrobial substances, such as antimicrobial peptides (bacteriocins), diacetyl, reuterin, hydrogen peroxide, and organic acids (lactic acid, propionic acid, acetic acid, benzoic acid) [14,15]. Any of these antimicrobial substances can be used in food production as partially purified compounds or can be delivered into food via antimicrobial-producing LAB cultures that are added as starter or adjunct cultures [16,17].

Traditional cheeses are a potential pool of new LAB strains with desirable biological and metabolic properties such as: acidogenic activity, production of antimicrobial compounds, production of proteinases, and probiotic properties [18,19,20]. Among LAB, the genera *Lactococcus* and *Enterococcus* are most commonly used as adjunct cultures in different products to control foodborne pathogens or to improve product quality [17,21,22,23]. However, to date, among many potential antimicrobial substances, only two bacteriocins (nisin and pediocin PA-1) have FDA approval and are commercially used in a variety of food products as natural preservatives [24,25].

In this study, LAB with potential antilisterial activity were isolated from traditionally made white brined cheeses. Their acidification activity and antibiotic susceptibility were determined in order to investigate their potential for application as components of starter or protective cultures in cheese production. According to their activities, one isolate was selected, identified and tested for its inhibitory activity against *L. monocytogenes* in UF cheese. Additionally, the efficacy of *Lactococcus lactis* subsp. *lactis* bv. *diacetylactis* BGBU1–4 in controlling *L. monocytogenes* growth in UF cheese was examined. This organism was previously isolated from the same cheese type and produces lactolisterin BU, a thermostable bacteriocin with antilisterial activity [26,27]. In a previous study, the strong antilisterial activity of *L. lactis* subsp. *lactis* BGBU1–4 in quark-type cheese was proven [28]. The aim of this study was to examine the effect of two LAB isolates exhibiting antilisterial activity on *L. monocytogenes* during storage of UF cheese.

## 2. Materials and Methods

### 2.1. Cheese Samples, Isolation of Lactic Acid Bacteria with Antilisterial Activity

White brined cheeses were produced in Sjenica, western Serbia, from unpasteurized cow’s milk and according to a traditional procedure without addition of starter cultures. Three cheese samples were taken from each cheese interior, individually homogenized using a sterile mortar and pestle, and 20 g was transferred to a stomacher bag with 180 mL of sodium citrate (2% *w*/*v*). The contents were homogenized in a Stomacher (Interlab, BagMixer 400P), and serial tenfold dilutions of the homogenates were prepared with sterile sodium chloride (0.85% *w*/*v*) and were surface plated onto two different growth media to isolate LAB: M17 agar (Merck GmbH, Darmstadt, Germany) supplemented with glucose (0.5% *w*/*v*; GM17) incubated at 30 °C and 37 °C for lactococci and enterococci, respectively, and de Man Rogosa and Sharp agar (MRS agar, Merck GmbH, Darmstadt, Germany) incubated on 30 °C and 37 °C for 24 h under aerobic conditions for lactobacilli. The plates with individual colonies were then overlaid with GM17 soft agar containing *L. monocytogenes* ATCC19111 and incubated for a further 24 h at 37 °C. Colonies with antilisterial activity were detected by the appearance of a zone of inhibition, and were purified using the medium on which they were originally cultivated. Antilisterial activity was confirmed by the agar well diffusion assay [29]. *L. monocytogenes* ATCC19111 (5 log cfu mL^−1^) was inoculated into GM17 soft agar, and wells were cut in the plates. Wells were filled with 50 µL of whole culture from an antilisterial producer (after 16 h incubation at appropriate temperature and medium) or cell-free supernatant (CFS). CFS of each whole culture was obtained by centrifugation (3500× *g*, room temperature, 10 min) and filtration of supernatant using 0.2 µm filter (Thermo Fisher Scientific, Waltham, MA, USA).

### 2.2. Proteinaceous Nature of Antilisterial Molecule

To determine the proteinaceous nature of antilisterial compounds, CFS and neutralized CFS (pH 7.00, adjusted with 1 M NaOH) were used. The test was performed by placing a crystal of proteolytic enzyme, pronase E (Sigma, St. Louis, MO, USA) close to the edge of the CFS-containing well; reduction of activity was taken as positive proteinaceous nature.

### 2.3. Identification of Lactic Acid Bacteria with Antilisterial Activity

Total DNA from antilisterial LAB isolates was obtained by the method of Hopfwood et al. (1986) [30] with minor modification. Bacterial cells grown in appropriate medium to early logarithmic (OD_600_ = 0.6–0.8) were collected by centrifugation (3500× *g*, room temperature, 10 min). Pellet was washed twice using TEN buffer (50 mM Tris-HCl pH 8; 10mM EDTA pH 8; 50 mM NaCl), further resuspended in solution of PP buffer (0.5 M saharoze; 40 mM NH4-acetata; 10 mM Mg-acetata; pH 7) with lysozyme (4 mg/mL) and incubated for 15 min at 37 °C. Identification of selected isolates was performed by 16s rRNA gene sequencing previously described by Golic et al. (2013) [27]. Platinum Taq DNA polymerase (Thermo Fisher Scientific, Waltham, MA, USA) was used to amplify the 16s rRNA gene using a GeneAmp PCR System 2700 thermal cycler (Applied Biosystems, Foster City, CA, USA) with specific primers, 16S—Fw (GAATCTTCCACAATGGACG) and 16S—Rev (TGACGGGCGGTGTGTACAAG) [31]. PCR products were visualized on 1% agarose gel at constant voltage of 80V. Excepted size of PCR products is 1500 bp. PCR products were purified using a Thermo Scientific PCR Purification Kit (Thermo Scientific, Lithuania) according to the manufacturer’s instructions. Purified PCR products were sequenced by the Macrogen Sequencing Service (Macrogen Europe, Amsterdam, The Netherlands). The BLAST algorithm was used for analyzing nucleotide sequences [32].

### 2.4. Detection of Hydrogen Peroxide

The ability of antilisterial isolates to produce H_2_O_2_ was confirmed using the method previously described by María Silvina Juárez Tomás et al. (2004) [33]. Solution A was prepared by mixing 12.5 mg 3,3’,5,5’-tetramethyl-benzidine (TMB; Sigma) and 3 mL of methanol (Merck, Darmstadt, Germany). Solution B was prepared with peroxidase (0.5 mg/mL, Sigma) and 1 mL sterilized mili-Q water. GM17 agar (10 mL held at 45 °C) was mixed with 0.6 mL of solution A and 0.2 mL of solution B, then poured into petri dishes to make TMB plates. Isolates were streaked on the surface of TMB plates and incubated at an appropriate temperature for 24 h. Colonies able to produce hydrogen peroxide turn blue or brown under ambient light, as H_2_O_2_ reacts with horseradish peroxidase in the agar to oxidize the TMB. All analyses of H_2_O_2_ production by antilisterial LAB were conducted in triplicate.

### 2.5. Detection of Diacetyl

Analyses of butanedione (diacetyl) in milk and antilisterial LAB culture were conducted using an Agilent 7890A gas chromatograph (GC) connected to an Agilent 7697A headspace device (HSS) and an electron capture detector (ECD), using the method previously described by Richelieu et al. (1997) [34]. Whole culture from an antilisterial producer (after 16 h incubation at appropriate temperature and medium) was washed twice in PBS buffer (0.1 M, pH 7) and finally resuspended in sterile reconstituted skim milk (10% *w*/*v*) (Nilac, The Netherlands). The samples were transferred to 20 mL glass headspace vials (5 mL of sample per vial) and hermetically sealed with crimped aluminum caps. High-purity nitrogen (>99.999%) was used to pressurize vials in the HSS device. The vials were equilibrated over 30 min at 70 °C. After equilibration, the samples were extracted using the single extraction mode, and transferred to the GC unit via the fused silica capillary transfer line (inner diameter 0.25 mm) at 100 °C loop temperature and 110 °C transfer line temperature. The GC was equipped with a Thermo Scientific™ TraceGOLD™ TG-5MT capillary column (60 m × 0.25 mm ID × 0.25 μm). For analyses of butanedione in the samples, the following oven temperature program was used: 45 °C for 2 min, then 10 °C/min to 150 °C, then hold at 150 °C for 27 min. The ECD operated at 250 °C with make-up gas flow of 30 mL/min. Butanedione in the samples was quantified using an external calibration method based on the concentration of the analyte in a standard series and the corresponding peak areas. For that purpose, butanedione analytical standard (97% purity; purchased from Sigma-Aldrich) was dissolved in water (milli Q ultrapure water) containing 4 ethanol (HPLC grade; purity ≥99.9%; purchased from Sigma-Aldrich). A standard series of butanedione solutions (0, 2, 5, 10, 15, 25, and 50 μg/L) was used to construct the calibration curve. All analyses were conducted in triplicate.

### 2.6. Antibiotic Resistance

The antibiotic resistance of selected isolates was determined by the Kirby-Bauer disk diffusion method, according to the Clinical and Laboratory Standards Institute [35]. Nine antibiotics were examined: streptomycin 300 µg (STR), ampicillin 10 µg (AMP), gentamicin 120 µg (GEN), vancomycin 30 µg (VAN), tetracycline 30 µg (TET), neomycin 30 µg (NEO), penicillin 10 U (PEN), erythromycin 15 µg (ERY), and chloramphenicol 30 µg (CHL). LAB isolate were classified as sensitive (S) or resistant (R) phenotype by the appearance of a zone of inhibition around antibiotic discs (BBL Sensi-Disc Antimicrobial Susceptibility Test Disc, Becton, Dickinson and Company, Franklin Lakes, NJ, USA).

### 2.7. Acidifying Activity

Acidifying activity of LAB isolates was examined using the International Dairy Federation Standard (IDF standard, 1995) [36]. The isolates were subcultured in appropriate medium (GM17 for lactococci or MRS for lactobacilli) and temperature (30 °C for lactococci or 37 °C for lactobacilli) for 16 h. Then, isolates were inoculated into sterile reconstituted skim milk (10% *w*/*v*) (Nilac, The Netherlands) at a level of 1 (*v*/*v*). The pH of milks was determined after 6, 12, and 24 h of incubation at the appropriate temperature. Analysis of acidifying activity of tested LAB isolates were done in triplicate.

### 2.8. UF Cheese Making Procedure

Control UF cheese (C) was made according to the procedure of Mazinani et al. (2014) [37] with minor modifications. Coagulation and fermentation was conducted at 30 °C during 17–18 h, and then 20 g/kg salt was added onto the cheese surface. Cheese was ripened at 12 °C during the next 7 days and stored at 4 °C for 35 days.

Experimental cheeses were produced using an identical procedure as for control cheese, except one LAB with antilisterial activity isolated in this study and/or *L. lactis* subsp. *lactis* BGBU1–4 was added to cheese at the same time as the starter culture. Those antilisterial LAB were previously labeled by streptomycin and rifampicin resistance, respectively, using the procedure described by Frece et al. (2005) [38], with minor modification. Both antilisterial LAB were cultured at appropriate conditions for 24h. The cultured cells of BGBU1–4 and antilisterial LAB from this study were added to plates containing 500 µg/mL of rifampicin (Sigma-Aldrich Chemie GmbH, Deisenhofen, Germany) 1 mg/mL of streptomycin, respectively, and incubated for 72 h at appropriate temperatures. Labeled antilisterial LAB were further used for cheese production. Overnight cultures were washed twice in PBS buffer (0.1 M, pH 7) and finally resuspended in the same buffer. The final suspensions were serially diluted and were added in suitable amounts to obtain 6 log cfu mL^−1^. At the same time, cheeses were artificially contaminated with *L. monocytogenes* ATCCC19111 (at three different contamination levels: ~3, 4, 5 log cfu mL^−1^), obtaining the 12 cheese variants presented in (Table 1). Control cheeses were produced with no added *L. monocytogenes* ATCCC19111 in order to obtain four additional variants (Table 1). All cheeses were made in triplicate.

### 2.9. Sampling and Microbiological Analysis of Experimental UF Cheeses

Experimental cheeses were analyzed five times during 35 days of storage at 4 °C: (i) immediately after ripening (day 0); (ii) after 7 days of storage; (iii) after 14 days of storage; (iv) after 21 days of storage; v) after 35 days of storage. Viable cell counts of antilisterial LAB and *L. monocytogenes* ATCC19111 were determined. Each experimental UF cheese was aseptically sampled (10 g), diluted in 90 mL sterile Ringer’s solution (0.85% *w*/*v*), and homogenized for 2 min in a Stomacher (Interlab, BagMixer 400P). After homogenization, tenfold dilutions were prepared for microbiological analysis. Antilisterial LAB isolated in this study was cultivated on plates containing 1 mg/mL streptomycin (Sigma-Aldrich Chemie GmbH, Deisenhofen, Germany) at appropriate conditions, while BGBU1–4_rif_ was cultivated on GM17 plates containing 500 µg/mL of rifampicin (Sigma-Aldrich Chemie GmbH, Deisenhofen, Germany) at 30 °C for 24 h. *L. monocytogenes* ATCC19111 was enumerated on Palcam Listeria selective agar base with Palcam Listeria selective supplement (Merck, Darmstadt, Germany), incubated at 37 °C for 48 h (ISO 11290-2:1998).

### 2.10. Statistical Analysis

All experiments were performed in triplicate. Two-way ANOVA was used to determine differences between experimental groups of milk and cheeses. The differences between means were compared using Student’s *t*-test, and were considered significant if *p* < 0.05.

## 3. Results and Discussion

### 3.1. Isolation and Identification of LAB with Antilisterial Activity

Among around 200 colonies isolated from cheeses, 20 were purified and selected according their zones of inhibition and antilisterial activity against *L. monocytogenes* ATCC19111. The percentage of colonies with antilisterial activity (among total LAB on the plates) was around 10%, which correlated with a previous study [39]. However, it differs from results previously reported by Campagnollo et al. [40], where up to 48.1% of tested bacterial colonies isolated from Minas cheese presented antilisterial activity. The reason for positive rate of antilisterial isolates could be a choice of type of cheese for isolation of bacteria, type of media used for isolation, type of antilisterial compound produced by isolates, and culture method. One isolate (PFMI565) had the strongest activity against *L. monocytogenes* ATCC19111 (zone of inhibition was 10 mm), and was chosen for further experiments. According to 16s rRNA gene sequencing, PFMI565 was identified as *Enterococcus durans*, showing 97.7% identity with many *Entorococcus durans* strains, including 4599, 4541, 4292, XT-1, and 3901. *Enterococcus* isolates can produce many antimicrobial substances including lactic acid, H_2_O_2_, bacteriocins, and bacteriocin-like substances (BLIS) [41,42,43,44]. The genera *Enterococcus* and *Listeria* are quite close according to molecular taxonomy and phylogenetic position, which could be a potential explanation for the antilisterial activity of enterococcoci [45,46,47].

### 3.2. Proteinaceous Nature of Antilisterial Compounds

Many strains of enterococci can produce bacteriocins, antimicrobial substances of protein nature [48]. To determine the nature of the antilisterial compound expressed by PFMI565, a crystal of pronase E was used in the agar well diffusion assay. After incubation, antilisterial activity was detected in the vicinity of the pronase E, indicating the antilisterial activity was not proteinaceous, so not bacteriocin or bacteriocin-like substances (Figure 1A). Similar results published previously indicated the antilisterial activity could be due to organic acid [43]. However, results from this study indicated that organic acid was not the only antilisterial compound, since although the zone of inhibition of neutralized CFS was smaller than that of CFS, it was clearly visible (Figure 1B).

### 3.3. Hydrogen Peroxide and Diacetyl Production

*E. durans* PFMI565 produced H_2_O_2_, as colonies turned blue. Previously, it has been shown that H_2_O_2_ produced and released by bacteria has the ability to inhibit other competent bacteria and host bacteria as well [49]. In some studies, LAB that are H_2_O_2_ producers inhibited the growth of foodborne and clinical pathogens such as *Staphylococcus aureus*, *Neiseria gonorrhea*, and *Gardnerella vaginalis* [50,51,52,53]. The antilisterial activity of neutralized CFS from *E. durans* PFMI565 is shown (Figure 1B), so the antilisterial activity of *E. durans* PFMI565 is likely to be due, at least in part, to H_2_O_2._

Diacetyl is a flavor-producing compound that can be produced by LAB and has antimicrobial activity against *L. monocytogenes* and *S. aureus* in liquid [54]. The diacetyl concentration of *E. durans* PFMI565 whole culture was 11.79 ± 0.44 µg/L, which corresponds to 0.01179 ppm. The minimum concentration of diacetyl required to achieve an antilisterial effect is 300 ppm [55]. In the specific conditions of packaging food in an atmosphere with 20% CO_2_, 50 ppm diacetyl showed antilisterial activity [56], which is still higher than the concentration detected in our pure culture of *E. durans* PFMI565. Comparing the results obtained in this study with others, it can be concluded that the concentration of diacetyl produced by *E. durans* PFMI565 was insufficient to achieve an antilisterial effect.

### 3.4. Antibiotic Resistance

Antibiotic resistance testing revealed *E. durans* PFMI565 was resistant to three out of nine antibiotics (gentamicin, streptomycin and neomycin), but was sensitive to six out of nine antibiotics (chloramphenicol ampicillin, vancomycin, tetracycline, penicillin and erythromycin). These results are partially in agreement with Amarel et al. 2017 [57], who showed that *E. durans* SJRP14, SJRP17 and SJRP26 were sensitive to clinically important antibiotics: erytromicin, tetracycline, kanamycin, penicillin, and vancomycin. Resistances to these antibiotics are usually a product of transformable genetic elements that are responsible for the transmission of antibiotic resistance determinants [57,58,59]. On the other hand, enterococci usually possess chromosomally encoded enzymes responsible for resistance to aminoglycosides (strepotmycin, gentamicin, neomycin), so transmission of these resistance genes is impossible [60]. Therefore, according to antibiotic susceptibility testing, the presence of *E. durans* PFMI565 in cheese as a starter or adjunct culture does not represent a risk for the spread of antibiotic resistance.

### 3.5. Acidifying Activity

In order to test the suitability of antilisterial strains *Lactococcus lactis* subsp. *lactis* BGBU1–4 and *Enterococcus durans* PFMI565 as starter culture components, acidifying activity was studied. The initial pH of skim milk was 6.58, and after 6 h of incubation with *L. lactis* subsp. *lactis* BGBU1–4 and *E. durans* PFMI565, pH was 5.82 (6 h) and 5.92 (6 h), respectively (Figure 2). After 24 h of incubation, pH of skim milk was reduced to 4.45 and 4.73 for *L. lactis* subsp. *lactis* BGBU1–4 and *E. durans* PFMI565, respectively. The pH of milk are statistically difference in the case of these two strains (Figure 2). Previously, it was reported that selected strains of *E. durans* and *L. lactis* subsp. *lactis* biovar *diacetylactis* can reduce the pH of skim milk to pH < 5, after 4 h of incubation [59,61,62], unlike the LAB used in this study. However, it is known that acidifying activity of LAB varies considerably [63]. A possible reason for differences in acidifying activity between strains of the same species could be related to levels of expression of ß-galactosidase or phosphor-ß-galastosidase [64]. Autochthonous LAB isolated from cheese could be used as a starter culture, important for standardization of production of autochthonous food products, as an adjunct culture for production of healthy and functional food product, or as a protective culture for prolonging shelf life [65]. Starter cultures used in cheese production and containing LAB usually can reduce milk pH to below 5.3 in 6 h [66]. According to the results of this study, *L. lactis* subsp. *lactis* BGBU1–4 and *E. durans* PFMI565 had only low acidifying activity in skim milk in the first 6 h of incubation, and therefore, they are not suitable to be used as starter cultures for cheese production. Nevertheless, *L. lactis* subsp. *lactis* BGBU1–4 and *E. durans* PFMI565 could be used as adjunct cultures in cheese and other dairy products, since both LAB showed antilisterial activity in vitro.

### 3.6. Microbiological Analysis of UF Cheeses

The ability of a lactolisterin BU producer (*L. lactis* subsp. *lactis* BGBU1–4) and a non-bacteriocin producer (*E. durans* PFMI565) that has antilisterial activity, to inhibit growth of *L. monocytogenes* in UF cheese, was studied. *L. lactis* subsp. *lactis* BGBU1–4 and *E. durans* PFMI565 were added during cheese production along with starter cultures, since both these LAB had average acidifying and strong antilisterial activities which are good attributes for adjunct cultures.

*L. monocytogenes* was not detected in control cheeses C, CB, C565, and B565C. In cheeses CL3, CL4 and CL5 produced with starter culture, without adjunct cultures BGBU1–4_rif_ and PFMI565_str_, and artificially contaminated with *L. monocytogenes* ATCC19111, numbers of this pathogen on the first day after ripening (day 0) were ~3.3 log cfu g^−1^, 4.4 log cfu g^−1^, and 5.5 log cfu g^−1^, respectively. At the end of storage, *L. monocytogenes* numbers dropped to ~2 log cfu g^−1^ in cheese CL3, ~2.2 log cfu g^−1^ in cheese CL4 and ~4.1 log cfu g^−1^ in cheese CL5. Results from previous studies indicate that *L. monocytogenes* decreased by more than 2 log cfu g^−1^ in soft feta-type cheeses during 90 days of storage at 4 °C [67,68]. This reduction in *L. monocytogenes* numbers during storage could be due to deleterious effects of fats and proteins on antilisterial molecules, influence of sodium chloride concentration and pH on activity of antilisterial molecules, or the artificially high initial level of inoculated *L. monocytogenes* [13,69]. Still, numbers of *L. monocytogenes* in all cheeses remained at levels that are not allowed by law in Serbia or in the EU (cheese must contain less than 10^2^ cfu g^−1^ (mL^−1^) [9,10].

Results of antimicrobial activity of strains BGBU1–4_rif_ and PFMI565_str_ against *L. monocytogenes* ATCC19111 in experimental cheeses are shown in (Figure 3). Statistical analysis has shown that both factors (type of strain and time of storage) and their interaction had significant effect on the number of *L. monocytogenes* ATCC19111 counted. In the control cheese variants, made without BGBU1–4_Rif_ and PFMI565_str_, statistically significant decrease in cells number of *L. monocytogenes* ATCC19111 was found during storage. However, the intensity of reduction rate was different, depending on whether cheese made with addition of BGBU1–4_rif_, PFMI565_str_ or their combination. The trend of a decreasing number of *L. monocytogenes* ATCC19111 did not depend on the initial number inoculated into cheeses. In cheeses with BGBU1–4_str_ and PFMI565_rif_, decreases in the number of *L. monocytogenes* followed different trends depending on whether cheese was produced with BGBU1–4_rif_, PFMI565_str_, or a combination of these two strains. The greatest reductions in *L. monocytogenes* numbers were achieved in cheese with BGBU1–4_str_ and in cheese with a combination of BGBU1–4_rif_ and PFMI565_str_. After 14 days of storage at 4 °C, *L. monocytogenes* counts in these two cheeses were decreased statistically significant, 1 log more than in cheese without antilisterial LAB. By the end of 21 days of storage, *L. monocytogenes* counts were statistically significant lower in all cheeses produced with antilisterial LAB than in control cheeses. In a previous study, it was shown that *L. lactis* subsp. *lactis* BGBU1–4 (a lactolisterin BU producer) has strong inhibitory activity against *L. monocytogenes* ATCC19111 in quark-type cheese during storage [28]. The results in our current study showed reductions of *L. monocytogenes* ATCC19111 in UF cheeses with *L. lactis* subsp. *lactis* BGBU1–4 and *E. durans* PFMI565 (separately) and in cheese with the combination of these two LAB, but still, viable *L. monocytogenes* remained in cheeses at detectable levels at the end of storage. Similar incomplete reductions of *L. monocytogenes* in cottage and quark-type cheese using nisin A, nisin Z, lacticin 481, and lactolisterin BU producers were also published [23,28]. The reason for this effect could be possible inactivation of bacteriocin or antimicrobial compound by its interaction with proteases or pH of cheese [70,71]. In some studies, it was also concluded that the control of *L. monocytogenes* in cheeses depends on LAB strain, nature of antimicrobial compounds and type of cheese [21,72].

Numbers of BGBU1–4_rif_ and PFMI565_str_ were also followed during 35 days of storage in all cheeses where they were used. After ripening, on the first day of storage (day 0), numbers of both LAB in all cheeses were ~8.30 log cfu g^−1^. Numbers of BGBU1–4_rif_ increased in cheeses during storage and reached ~8.75 cfu g^−1^ after 35 days of storage. On the other hand, numbers of PFMI565_str_ decreased somewhat and were ~7.5 log cfu g^−1^ after 35 days of storage. In general, both examined LAB showed good ability to survive in UF cheese during storage. These results confirm previous findings indicating very good survival of enterococci and BGBU1–4 in cheese during storage [21,22,29,40]. In both cases, numbers of BGBU1–4_rif_ and PFMI565_str_ did not depend on the initial number of *L. monocytogenes*. Numbers of BGBU1–4_rif_ were higher than numbers of PFMI565_str_ at the end of storage, which could be due to the stronger antilisterial effect in of lactolisterin BU producer compared with *E. durans* PFMI565 in the UF cheese. Our study confirmed in vitro results [26] (Figure 1A,B) and showed that BGBU1–4_rif_ and PFMI565_str_ produce antilisterial effects in UF cheese during 35 days of storage at 4 °C.

## 4. Conclusions

In total, 20 LAB colonies with antilisterial activity were isolated and purified from Serbian white cheese. Among the 20 LAB, the isolate with the strongest antilisterial activity was identified as *Enterococcus durans* PFMI565.

*E. durans* PFMI565 produces little acidifying activity in milk and is sensitive to clinically important antibiotics, making it a good candidate for application in cheese production as an adjunct culture. Furthermore, addition of *E. durans* PFMI565 and *Lactococcus lactis* susp *lactis* BGBU1–4 (bacteriocin producer) into UF cheeses artificially contaminated with *Listeria monocytogenes* resulted in antilisterial effects during 35 days of storage. The reduction of *L. monocytogenes* is significantly greater in UF cheeses made with *L. lactis* susp *lactis* BGBU1–4 and in cheese with a combination of both *L. lactis* susp *lactis* BGBU1–4 and *E. durans* PFMI565.

The findings in this study indicate that the autochthonous LAB, *E. durans* PFMI565 and *L. lactis* subsp. *lactis* BGBU1–4, used as protective cultures in production of UF cheeses, would provide protection against growth of *L. monocytogenes* in UF cheese. However, the nature of the antilisterial compound(s) produced by *E. durans* PFMI565 is not yet resolved, although bacteriocin has been ruled out. Additionally, the presence of virulence factors in *E. durans* PFMI565 must be determined. Therefore, additional investigation of *E. durans* PFMI565 is required before this LAB could be used in any applications for commercial UF cheese production.

## Figures and Tables

**Figure 1 foods-10-01448-f001:**
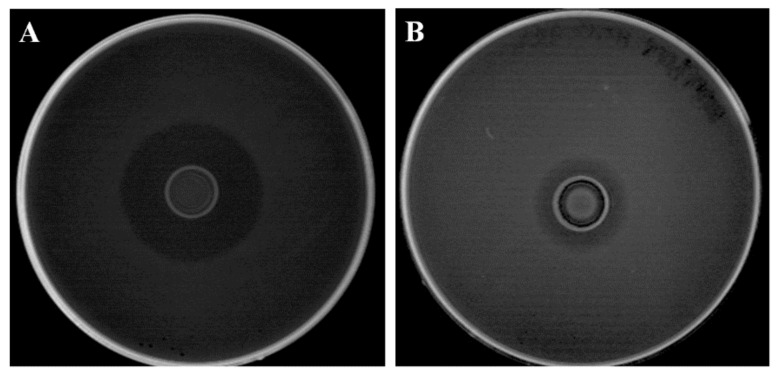
Antilisterial activity of *Enterococcus durans* PFMI565 against *Listeria monocytogenes* ATCC19111; (**A**) Cell-free supernatant (CFS) of *E. durans* PFMI565; (**B**) Neutralized CFS (pH 7.00) of *E. durans* PFMI565.

**Figure 2 foods-10-01448-f002:**
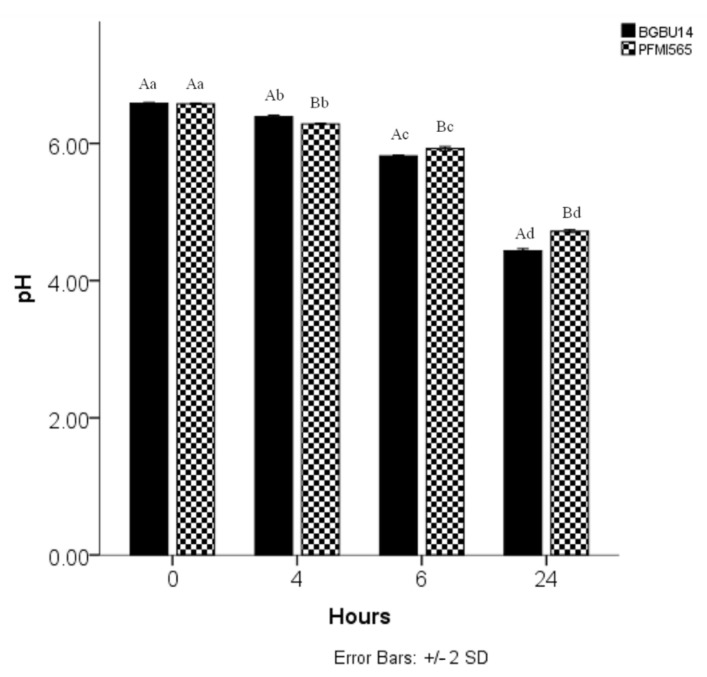
Acidifying activity of *Lactococcus lactis* subsp. *lactis* BGBU1–4 and *Enterococcus durans* PFMI565 in skim milk during 24 h of incubation at 30 °C. Values represent mean value +/– standard deviation (*n* = 3). Small letter indicated statistical significant difference between pH of milk of same strain during storage. Big letter indicate statistical significant difference between pH of milk with *L. lactis* subsp *lactis* BGBU1–4 and *E. durans* PFMI565 in same hour of incubation.

**Figure 3 foods-10-01448-f003:**
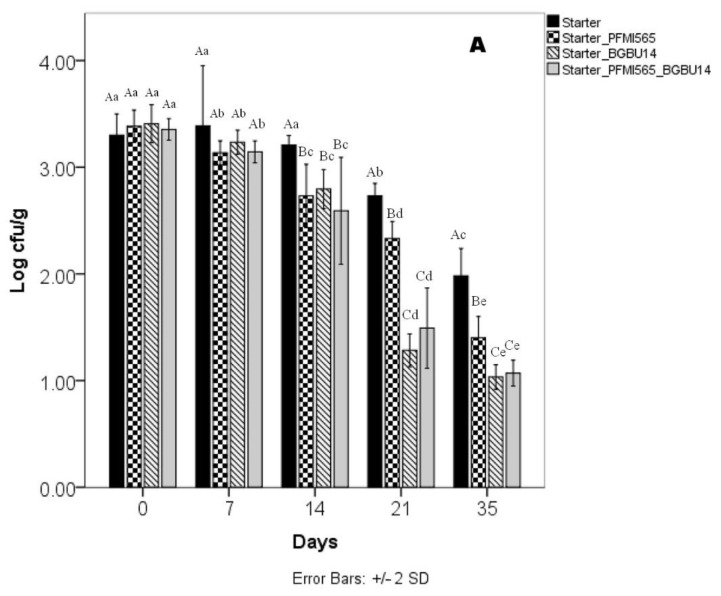
The effect of *Lactococcus lactis* subsp. *lactis* BGBU1–4 and *Enterococcus durans* PFMI565 on survival of *Listeria monocytogenes* ATCC19111 in UF cheeses with (**A**) Initial number of *L. monocytogenes* ATCC19111 ~3 log cfu g^−1^; (**B**) Initial number of *L. monocytogenes* ATCC19111 ~4 log cfu g^−1^; (**C**) Initial number of *L. monocytogenes* ATCC19111 ~5 log cfu g^−1^. Bars represent means +/– standard deviations (*n* = 3). Same small letter indicated there is no statistical significant difference in cell number of *L. monocytogenes* ATCC1911 in same sample during storage. Big letter indicate statistically significant differences in cell number of *L. monocytogenes* ATCC19111 between different samples at the same day of storage.

**Table 1 foods-10-01448-t001:** Starter and adjunct culture used for UF cheese production and level of *Listeria monocytogenes* ATCC19111 contamination.

Cheese Designation	Bacterial Cultures and Level of Contamination
C	CHN 11
CB	CHN11, BGBU1–4
CBL3	CHN11, BGBU1–4, *L.monocyt*. 3 log cfu g^−1^
CBL4	CHN11, BGBU1–4, *L.monocyt.* 4 log cfu g^−1^
CBL5	CHN11, BGBU1–4, *L.monocyt*. 5 log cfu g^−1^
C565	CHN11, isolate-PFMIX*
C565L3	CHN11, isolate-PFMIX**, L.monocyt*. 3log cfu g^−1^
C565L4	CHN11, isolate-PFMIX*, *L.monocyt.* 4 log cfu g^−1^
C565L5	CHN11, isolate-PFMIX*, *L.monocyt.* 5 log cfu g^−1^
B565C	CHN11, BGBU1–4, isolate-PFMIX*
B565CL3	CHN11, BGBU1–4, isolate-PFMIX*, *L.monocyt*. 3 log cfu g^−1^
B565CL4	CHN11, BGBU1–4, isolate-PFMIX*, *L.monocyt.* 4 log cfu g^−1^
B565CL5	CHN11, BGBU1–4, isolate-PFMIX*, *L.monocyt.* 5 log cfu g^−1^

Isolate-PFMIX*—isolate with the strongest activity against *Listeria monocytogenes* ATCC19111 obtained in this study.

## Data Availability

Data available in a publicly accessible repository.

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
