# Peer review of "Autochthonous Enterococcus durans PFMI565 and Lactococcus lactis subsp. lactis BGBU1–4 in Bio-Control of Listeria monocytogenes in Ultrafiltered Cheese"

_foods, 2021, doi:10.3390/foods10071448_

Round 1

Reviewer 1 Report

REVIEWER COMMENTS TO AUTHORS:

Generally, this is a clear, concise, and well-written manuscript on a relevant topic.  Adequate information about previous findings is described for readers to follow and understand the procedures and rationale behind this present study. Overall, the methods are appropriate. However, clarification of a few details and rationales should be provided. Apart from some minor concerns, the results are clear and well-written.  The authors have offered a systematic contribution to the research literature in this area. Overall, the findings in this manuscript have implications for the theoretical basis, development, and production of UF cheeses using Enterococcus durans PFMI565 and Lactococcus 2 lactis subsp. lactis BGBU1-4 as protective cultures. Specific comments follow.

INTRODUCTION

  1. 1, lines 31 - 32: The statement should be rewritten as “Cheeses are a significant part of human diets because of their chemical composition and high contents of vitamins, fatty acids, minerals, bioactive compounds, and probiotic bacteria.” 
  2. 2, lines 54 - 56: Please provide reference(s) for the first statement.
  3. 2, line 64: The word "biological" or “biotechnological” would be a better word choice than the word, "technological."

MATERIALS/METHODS

  1. 3, line 94: Consider providing the full name of the MRS agar.
  2. 3, lines 114-115: Kindly state the rationale and also describe the minor modification made to pretreatment with lysozyme.
  3. 4, lines 175 -176: The authors should consider providing the name of the medium used in the subculturing of the isolates.
  4. 4, lines 181-182: Kindly state the rationale and also describe the minor modification made in the preparation of the control UF cheese.
  5. 5, lines 195 -197: Kindly state the rationale and also describe the minor modification made in labeling the antilisterial LAB by streptomycin and rifampicin resistance.

RESULTS/DISCUSSION

Results of the statistical analysis were not mentioned in the results/discussion.

CONCLUSION

Well written.

REFERENCE

Be consistent with your pattern of reference. i.e provide the missing DOI for some of the references.

Reviewer 2 Report

The manuscript is extremely interesting and important for food technology, because its topic is testing new strains with potential anti-listeria and pro-health properties.

The paper is well organized and well elaborated, minor revisions mentioned below:

  • Please, provide information on the number of repetitions for each chemical analysis described in "2. Materials and Methods" point
  • What kind of milk was used to obtain the cheese described under point " 2.1. Cheese samples, isolation of lactic acid bacteria with antilisterial activity"? Why for the determination of lactobacilli the Petri dishes were incubated in 37 ° C instead of 30 ° C?
  • Have the interactions between the FD-DVS CHN-11 primer bacteria and the anti-listeria tested strains been studied? This information should be included in “2.8. UF cheese making procedure” point, in my opinion
  • Figure 2 and Figure 3 should present lines, not bars, because time (put on the x-axis) is a continuous parameter
  • Why the manuscript does not present the results of the statistical analysis?

Reviewer 3 Report

Dear authors

The design of this study is absolutely appropriate.  This is the way we do it, searching for new bacterial species with properties that can be used in food production. The use of these indigenous species is widespread. 

Check font size throughout the manuscript.

Check  "Foods" guidelines for manuscripts.

The introduction is sufficient, however a spelling mistake in

line 33 ultrafiltration

The materials and method section describes most of the methods used. In this study you are using a strain (L. lactis BGBU1-4) from an earlier study. In the last experiment this strain is made resistant to rifampicin. This must be described in this section. 

line 117 16S rRNA-gene (non cursive)

line 121 16S rRNA-gene primers  are widely used for bacterial identification. What is the size of the fragment?

line 193 results from the study is mentioned in the method section

line 205 I have problems with table 1. In this table, the results of this study for an indigenous LAB described in the table.  It needs to be anonymized and described e.g  " x-the isolated from the screening with the best characteristics" 

line 214, 215 Results form the study is mentioned in the method section This must be described differently If the LAB isolate you wanted to study further not was resistant to any known antibiotics, what is the procedure then?

Figure 1  It is not very clear to me what the result is here. Improve.

line 280 streptomycin is here mentioned twice while chloramphenicol is missing

Figure 3 I am not convinced that the result of reduced viable cell count of Listeria is due to the antilisteral activity of the LAB isolates or the fact that L. monocytogenes is weak to competition?  What is the total viable count? 

Some of the references are missing author names or other identifications (5, 9, 10, 22, 37). The use of fonts in the reference list is not consistent.

Success with your manuscript!

Round 2

Reviewer 2 Report

I can see that the authors corrected the manuscript taking into account the comments of the reviewers. However, not everything has been completed and corrected properly.

  • In section "2.3. Identification of lactic acid bacteria with antilisterial activity" please indicate centrifugation speed as "g" instead of "rpm".
  • In this section you also need to correct the word "saharose"